# Assessment of the Activity of Decoquinate and Its Quinoline-*O*-Carbamate Derivatives against *Toxoplasma gondii* In Vitro and in Pregnant Mice Infected with *T. gondii* Oocysts

**DOI:** 10.3390/molecules26216393

**Published:** 2021-10-22

**Authors:** Jessica Ramseier, Dennis Imhof, Nicoleta Anghel, Kai Hänggeli, Richard M. Beteck, Vreni Balmer, Luis-Miguel Ortega-Mora, Roberto Sanchez-Sanchez, Ignacio Ferre, Richard K. Haynes, Andrew Hemphill

**Affiliations:** 1Institute of Parasitology, Vetsuisse Faculty, Department of Infectious Diseases and Pathobiology, University of Bern, Länggass-Strasse 122, 3012 Bern, Switzerland; dennis.imhof@vetsuisse.unibe.ch (D.I.); nicoleta.anghel@vetsuisse.unibe.ch (N.A.); kai.haenggeli@vetsuisse.unibe.ch (K.H.); vreni.balmer@vetsuisse.unibe.ch (V.B.); 2Graduate School for Cellular and Biomedical Sciences, University of Bern, Mittelstrasse 43, 3012 Bern, Switzerland; 3Centre of Excellence for Pharmaceutical Sciences, Faculty of Health Sciences, North-West University, Potchefstroom 2520, South Africa; 25159194@nwu.ac.za (R.M.B.); haynes@ust.hk (R.K.H.); 4SALUVET, Animal Health Department, Faculty of Veterinary Sciences, Complutense University of Madrid, Ciudad Universitaria s/n, 28040 Madrid, Spain; luis.ortega@ucm.es (L.-M.O.-M.); robers01@ucm.es (R.S.-S.); iferrepe@vet.ucm.es (I.F.)

**Keywords:** quinolone, decoquinate, *Toxoplasma gondii*, tachyzoites, bradyzoites, oocysts, vertical transmission, proliferation, resistance

## Abstract

The quinolone decoquinate (DCQ) is widely used in veterinary practice for the treatment of bacterial and parasitic infections, most notably, coccidiosis in poultry and in ruminants. We have investigated the effects of treatment of *Toxoplasma gondii* in infected human foreskin fibroblasts (HFF) with DCQ. This induced distinct alterations in the parasite mitochondrion within 24 h, which persisted even after long-term (500 nM, 52 days) treatment, although there was no parasiticidal effect. Based on the low half-maximal effective concentration (IC_50_) of 1.1 nM and the high selectivity index of >5000, the efficacy of oral treatment of pregnant mice experimentally infected with *T. gondii* oocysts with DCQ at 10 mg/kg/day for 5 days was assessed. However, the treatment had detrimental effects, induced higher neonatal mortality than *T. gondii* infection alone, and did not prevent vertical transmission. Thus, three quinoline-*O*-carbamate derivatives of DCQ, anticipated to have better physicochemical properties than DCQ, were assessed in vitro. One such compound, RMB060, displayed an exceedingly low IC_50_ of 0.07 nM, when applied concomitantly with the infection of host cells and had no impact on HFF viability at 10 µM. As was the case for DCQ, RMB060 treatment resulted in the alteration of the mitochondrial matrix and loss of cristae, but the changes became apparent at just 6 h after the commencement of treatment. After 48 h, RMB060 induced the expression of the bradyzoite antigen BAG1, but TEM did not reveal any other features reminiscent of bradyzoites. The exposure of infected cultures to 300 nM RMB060 for 52 days did not result in the complete killing of all tachyzoites, although mitochondria remained ultrastructurally damaged and there was a slower proliferation rate. The treatment of mice infected with *T. gondii* oocysts with RMB060 did reduce parasite burden in non-pregnant mice and dams, but vertical transmission to pups could not be prevented.

## 1. Introduction

*Toxoplasma gondii*, one of the most successful parasites on this planet, infects a wide variety of mammalian species and almost one-third of the human population. Felids are the definitive hosts, whereas virtually all warm-blooded animals, including humans, serve as intermediate hosts [1]. The life cycle of the parasite comprises three distinct stages: (i) tachyzoites, which are rapidly proliferating and cause acute disease by undergoing continuous host cell invasion, proliferation and egress (lytic cycle); (ii) bradyzoites, which divide slowly, form tissue cysts and can persist within intermediate hosts for extended periods of time; (iii) sporozoites, encapsulated in sporocysts within sporulated oocysts. Sexual development takes place in the intestinal tissue of felids, resulting in the formation of oocysts, which are shed into the environment. Infection takes place through the oral route, either by the ingestion of tissue cysts in raw or undercooked meat or by oocysts present in the environment, and also through the vertical transmission of tachyzoites upon primary infection during pregnancy [2]. In ruminants, such as sheep and goats, and other farm animals, toxoplasmosis often causes abortion, which has serious economic consequences. In humans, most infections are asymptomatic, but toxoplasmosis can inflict serious disease in immunocompromised individuals and newborns. The high zoonotic potential of this disease poses a serious health concern worldwide [3].

The drugs currently used for the chemotherapy of human and animal toxoplasmosis are either sulfadiazine or clindamycin, both combined with pyrimethamine, or trimethoprim-sulfamethoxazole, with leucovorin added to prevent hematologic toxicity. Atovaquone or azithromycin, combined with pyrimethamine or sulfadiazine, are alternative treatment options [4]. The macrolide antibiotic spiramycin is prescribed immediately after the diagnosis of maternal infection and changed to pyrimethamine-sulfonamide combination when fetal infection is diagnosed or in the case where infection is acquired in late pregnancy [5]. However, treatment failures caused by the need to cease treatment due to serious adverse events are frequently observed, and the prolonged courses of therapy required for treatment and the suppression of infection are a serious problem. In addition, drug resistance can occur by as yet unspecified pathways [6]. The currently chemotherapeutic treatment regimens affect the acute stage of toxoplasmosis, but will not completely eradicate the infection during the active multiplication of the parasite [7], thus leaving an escape route for tachyzoites and enabling them to differentiate into bradyzoites, which form long-lived and drug-resistant tissue cysts. There is no *Toxoplasma* vaccine available for use in humans, but a live-vaccine (Toxovax^R^, MSD Animal Health) that has been on the market for many years helps to prevent abortion in sheep flocks [1]. Thus, there is an urgent need for the development of new safe and effective drugs for the treatment of toxoplasmosis. A promising approach for the identification of novel chemotherapeutic options is the repurposing of drugs that were originally developed for other indications [8,9,10].

Decoquinate (DCQ; ethyl 6-decyloxy-7-ethoxy-4-oxo-1*H*-quinoline-3-carboxylate, Figure 1A) is a quinolone derivative that was originally developed for the treatment of coccidiosis in poultry, but is now also commercialized in many countries worldwide for use in ruminants [11]. DCQ inhibits oxidative phosphorylation and, thus, the generation of ATP in the mitochondrion by interfering with the electron transfer from ubiquinone to cytochrome *c*. It, thereby, acts as a cytochrome *bc_1_* inhibitor [12]. DCQ has a good safety profile, with an LD_50_ in rats of >5000 mg/kg and is used as a coccidiostat by adding to animal feed (see https://www.drugs.com/pro/deccox.html) (accessed on 15 October 2021). Although it is highly lipophilic and has exceedingly low aqueous solubility (0.06 μg/mL), it is able to be used for the treatment of intestinal infections [13]. DCQ has been formulated to treat gastrointestinal coccidiosis in cattle, small ruminants and poultry at concentrations of approximately 0.5 mg/kg body weight. In milking cows treated with the formulated DCQ, the maximum concentration of DCQ attained in plasma was 2 µM, a level well above its therapeutic effect [14].

The efficacy of DCQ against *T. gondii* in vitro has been demonstrated; it inhibits *T. gondii* proliferation in vitro with an IC_50_ value of 12 nM [15]. DCQ also has prophylactic properties. In ewes that were submitted to treatment with DCQ commencing 10 days prior to infection with *T. gondii* oocysts, treatment resulted in decreased pathological changes in the placental tissue, increased viability and weight of newborn lambs, and reduced febrile and humoral immune responses upon infection [16].

In the light of the notoriously poor drug-like properties of DCQ, attempts have been made to improve these properties. Thus, highly polar groups have been added to generate DCQ prodrugs with enhanced water solubility [17]. In our case, in order to render DCQ more useful for treating systemic infections, DCQ was converted into the quinoline-*O*-carbamate derivatives RMB054, RMB055 and RMB060 (Figure 1). These quinoline-*O*-carbamates displayed in vitro efficacies with half maximal effective concentrations (IC_50_ values) against *T. gondii* in the low nM range, and negligible cytotoxicity against human foreskin fibroblasts and human fetal lung fibroblasts [18].

In this study, we demonstrate the efficacies of RMB054, RMB055 and RMB060 in comparison with DCQ in vitro, describe the ultrastructural changes upon drug exposure, and investigate the effects of long-term treatments with DCQ and the derivatives in vitro. In addition, we assessed the efficacy of RMB060 against *T. gondii* oocyst infection in pregnant mice in comparison with DCQ.

## 2. Results

### 2.1. Efficacy of DCQ against T. gondii In Vitro and in CD1 Mice Infected with T. gondii Oocysts

The efficacy of DCQ in vitro was determined using *T. gondii* β-gal constitutively expressing β-galactosidase grown in HFF monolayers. When DCQ was added concomitantly with infection, the IC_50_ was 1.1 nM. As DCQ did not affect HFF viability at 5 µM, it thus displays a selectivity index of >5000. The efficacy was in a similar range (IC_50_ = 4.7 nM) when the compound was added to cultures already harboring intracellular parasites (see Table 1; Appendix A). The low IC_50_ values and the low degree of cytotoxicity against HFF host cells in vitro encouraged us to examine the efficacy of oral treatment of CD1 mice infected with *T. gondii* oocysts with DCQ at 10 mg/kg/day for 5 days. The results of this experiment are summarized in Table 2. 

One dam of the DCQ group showed clinical signs of oocyst infection, including ruffled coat, weak appearance and respiratory problems, and had to be euthanized one day after she gave birth. All pups were born dead and, during necropsy, four more pups were found. Furthermore, two dams from the DCQ and one from the C+ group gave birth to non-viable pups. Overall, DCQ treatment led to increased neonatal pup mortality of 70% (64 out of 92 pups) compared to 40% (37 out of 92 pups) in the non-treated but infected C+ group (Figure 2A). In addition, 3 of the 28 remaining pups from the DCQ group died during the first 16 days of the 1-month follow-up phase, while no postnatal mortality was observed in the C+ group (Table 2). Of the 25 surviving pups in the DCQ group, all were tested PCR-positive for *T. gondii* in the brain, and in the C+ group 35 of the 55 surviving pups (64%) were tested PCR-positive. In the adult mice, including dams and non-pregnant mice, of the DCQ and the C+ group, brain infection was detected in 11 out of 12 mice (Table 2). No difference in parasite load between the DCQ and the C+ group could be observed, and the mean values were even slightly higher in the DCQ group (Figure 2B,C). In all *T. gondii* brain positive mice, increased IgG serum-levels were recorded, and IgG titers of the DCQ and the C+ group were in the same range (Appendix A). Thus, despite highly promising IC_50_ values and the low cytotoxicity seen in vitro, DCQ treatment did not protect adult mice from *T. gondii* infection and had a detrimental impact on pup survival by causing more neonatal mortalities than those infected with *T. gondii* alone.

### 2.2. Efficacies of the Quinoline-O-Carbamate Derivatives of DCQ In Vitro

Due to the negative effects of treatment with DCQ, we investigated the efficacy in vitro of the three DCQ derivatives RMB054, RMB055 and RMB060 (Figure 1) [18] in comparison to DCQ. Different concentrations of DCQ and the derivatives in DMSO were added to HFF either concomitantly with infection of *T. gondii-*β-gal tachyzoites, or 3 h after infection. In the former case, IC_50_ values were 11.5 nM for RMB054, 199.6 nM for RMB055, and 0.07 nM for RMB060. When compounds were added after infection, the IC_50_ values for RMB054 and RMB055 remained in a similar range, while it was tenfold inferior for RMB060, indicating that the last compound not only had an impact on intracellular proliferation but also exerted pronounced activity on extracellular tachyzoites and inhibited host cell invasion. The corresponding dose-response curves are shown in Appendix A. The viability of uninfected HFF was not impaired by the compounds up to concentrations of 5 µM (DCQ and RMB054) or 10 µM (RMB055, RMB060) (Table 1).

Due to its excellent selectivity and efficacy, RMB060 was chosen for further investigation. The alterations in parasite ultrastructure induced by these compounds were studied by TEM. *T. gondii* Me49 tachyzoites grown in absence of compound (Figure 3A,B) exhibited typical features of apicomplexan parasites. They were situated in a parasitophorous vacuole, separated from the host cell cytoplasm by a parasitophorous vacuole membrane. The conoid at the apical end, and secretory organelles, such as micronemes, rhoptries and dense granules were clearly discernible, and tachyzoites divided by endodyogeny. *T. gondii* has a single mitochondrion, which exhibits a tubule-like structure, of which only portions are visible on a single section plane. The mitochondrion is characterized by an electron-dense matrix and numerous cristae, all of which forms a clearly discernible organelle (Figure 3A,B).

After treatment with 500 nM RMB060 for 6 h, the mitochondrial matrix had a less electron dense appearance and dissolved cristae became evident, while the host cell mitochondria remained structurally intact (Figure 3C). Treatment for 24 h led to a further deterioration of the mitochondrial matrix which appeared increasingly electron-lucent, but the outer membrane remained structurally intact, and membranous and particulate material accumulated at the interior of this organelle (Figure 3D). Apparently, dividing parasites were visible in many instances, although there were mostly not more than two tachyzoites per vacuole. After 48 h (Figure 3E) and 72 h (Figure 3F), the mitochondrial matrix of tachyzoites appeared more electron dense and cristae became more evident again. In parallel, the parasite cytoplasm became increasingly more vacuolized, and these vacuoles were either empty or filled with particulate and often electron dense material of unknown nature. Parasitophorous vacuoles harboring several parasites were now found, especially in the specimens fixed after 72 h of drug treatment, indicating that tachyzoites had resumed proliferation, albeit at a much slower rate. In cultures treated with DCQ, RMB054 or RMB055, similar results could be observed (Appendix A). 

Under unfavorable culture conditions, *T. gondii* tachyzoites can, in some instances, induce differentiation and undergo increased bradyzoite gene expression. To investigate whether attenuated proliferation of RMB060 treated parasites is accompanied by the expression of BAG1 (bradyzoite antigen 1), a heat shock protein and marker for physiological stress that is highly expressed in *T. gondii* bradyzoites, immunofluorescence staining of drug treated *T. gondii* Me49 in vitro cultures was performed (Figure 4). 

For this, a low-passage number of *T. gondii* strain (TgPgSp1) with high BAG1 expression without drug treatment was used as a control, and BAG1 staining had a cytoplasmic distribution within these tachyzoites. The maintenance of *T. gondii* Me49 in the absence of treatment did not result in BAG1 expression, but treatment with RMB060 for 48 h resulted in the expression of the bradyzoite marker BAG1 in the cytoplasm of some, but not all, parasites (Figure 4). DCQ also induced the expression of BAG1 after 48 h (Figure 4), but in RMB054 treated cultures, BAG1 expression was not clearly detectable (Appendix A). Despite the expression of the bradyzoite marker BAG1, it was not possible to identify by TEM any other bradyzoite-like features (e.g., tissue cyst wall formation, amylopectin granules), either in cultures that had undergone long-term treatments with RMB060 (Figure 5A,B), or in tachyzoites treated with DCQ for 24 h or for extended periods of time (Figure 5C,D). For both compounds, however, we found that the structural alterations seen in the mitochondrion were still evident in tachyzoites that had undergone long-term treatments.

### 2.3. Safety Assessment of DCQ and RMB060 during Pregnancy in BALB/c Mice

Due to the increased neonatal mortality observed in DCQ-treated pregnant mice infected with *T. gondii*, potential safety issues, such as drug-induced interference in pregnancy, were assessed in non-infected BALB/c mice. The mice mated and received 10 mg/kg/day DCQ or RMB060 emulsified in corn oil on days 9–13 post-mating. In the control group treated with corn oil only, four out of six mice became pregnant and gave birth to 19 pups, of which 1 died within the first 2 days post-partum (neonatal mortality). In the DCQ-treated group, only two out of six mice became pregnant, and in total only seven pups were born alive, which survived until the end of the assessment. In the RMB060 group, four out of six mice became pregnant with 27 pups in total, of which three died within the first two days after birth. This implies that there is a strong interference in pregnancy outcome with a greatly reduced number of newborn pups, due to DCQ treatment, but only minor interference in pregnancy due to RMB060 treatment (Table 3). 

### 2.4. Efficacy of RMB060 in CD1 Mice Infected with T. gondii Oocysts

Due to the notably potent effect of RMB060 in vitro, as reflected by the very low IC_50_ values and the low degree of cytotoxicity against HFF host cells, we examined the efficacy of oral RMB060 treatment (10 mg/kg/day for 5 days) in CD1 mice infected with *T. gondii* oocysts. The results of this experiment are summarized in Table 4.

One dam of the RMB060 group gave birth to non-viable pups and one dam of the C+ group was found dead in the cage before she could give birth. During necropsy, eight pups were found in the uterus, but the reason for the death of the dam was not elucidated. Overall, the treatment with RMB060 led to a slight, but non-significant, decrease in neonatal mortality (16.8%; 21 out of 125 pups) compared to 23% (26 out of 113 pups) in the non-treated but infected C+ group (Figure 6A). In the RMB060 group, four out of the 104 remaining pups (3.8%) died during the first 11 days of the postnatal phase, while eight out of the remaining 87 pups (9.2%) died in the C+ group (Table 4). Of the 100 surviving pups in the RMB060 group, 62 were tested PCR-positive for *T. gondii* in the brain (62%), and in the C+ group 70 of the 79 surviving pups (89%) were tested PCR-positive. In the adult mice (including dams and non-pregnant mice), brain infection was detected in eight out of 12 mice in the RMB060 group and in 11 out of 12 mice in the C+ group (Table 4). The mean values of the parasite load were slightly higher in the C+ group compared to the RMB060 group in dams and non-pregnant mice, but there was no significant difference (Figure 6B,C). 

In all *T. gondii* brain positive mice, increased IgG serum-levels were recorded. In non-pregnant mice, IgG titers of the C+ group showed higher RIPC values than those of the RMB060 group, but in dams the IgG titer of both groups were in the same range (Appendix A). Thus, despite highly promising IC_50_ values and the low cytotoxicity seen in vitro, RMB060 treatment did not protect all dams from *T. gondii* infection and only a low increase in pup survival compared to the C+ group could be achieved.

## 3. Discussion

Quinolones are a group of compounds that are active against a wide array of bacteria and parasites [19]. They are in clinical use as antibiotics, and the anti-malarial activity of these compounds has been known for many years [20,21,22,23,24]. Quinolones also possess activity against *T. gondii*, and, in particular, some endochin-like quinolones—applied alone or in combination with pyrimethamine—were shown to exhibit promising effects against acute and also latent toxoplasmosis [25,26,27]. DCQ is one of the most prominent and widely used quinolones. It is frequently applied in chicken and ruminants against coccidiosis, caused by different *Eimeria* species that colonize and cause damage in the gastrointestinal tract. A drawback of DCQ is its exceedingly poor aqueous solubility, which renders the compound less useful for combating tissue-dwelling parasites. 

In this study, we investigate the efficacies of DCQ and three quinoline-*O*-carbamate derivatives RMB055, RMB054 and RMB060 against *T. gondii* tachyzoites. All three derivatives were prepared from DCQ in an attempt to improve the drug-likeness of the DCQ scaffold by adding groups that improve the physicochemical properties of the drug [18]. RMB060 was highly active against the liver stage of *Plasmodium berghei* malaria parasites and all three derivatives were also highly active against multi-drug-resistant strains of *P. falciparum*, with no or very minor cytotoxicity reported in human fetal lung fibroblasts and human foreskin fibroblasts [18].

DCQ was reported previously to inhibit *T. gondii* proliferation in vitro with an IC_50_ value of 12 nM. Similar to other quinolones, the drug affects the cytochrome *bc_1_* complex-mediated electron transport in the mitochondrion [15]. The cytochrome *bc_1_*-complex transfers electrons from coenzyme Q to cytochrome *c* [28]. DCQ blocks this transfer, thereby causing the relocation of electrons to other biomolecules generating free radicals and reactive oxygen species (ROS), which are toxic to the parasite. More specifically, DCQ blocks the quinol reductase site of the parasite mitochondrial cytochrome *bc_1_* complex and inhibits parasite mitochondrial electron transport chain [29,30].

In our study, the IC_50_ for DCQ was found to range from 1.1–4.7 nM, depending on the time point of initiation of treatment. Thus, we confirmed previous findings on *T. gondii,* and DCQ had also been reported to be active against two closely related apicomplexan parasites, *Besnoitia besnoiti* and *Neospora caninum* [12,15,31]. Against the apicomplexan *Sarcocystis neurona*, DCQ displayed IC_50_ values for merozoites in the range of 0.6–1.1 nM [32], and had parasiticidal activity against *S. neurona* schizonts on a 10 day treatment at 240 nM. Notably, long-term treatment assays in our study showed that DCQ, applied at concentrations of up to 500 nM per day for 24 days was not parasiticidal. Whilst a report of an earlier study of treatment of infected cultures with 240 nM DCQ led to tissue cyst formation of the *T. gondii* RH strain [33], we were not able to detect any structural features associated with formation of bradyzoites by TEM. Whilst the use of different strains, culture conditions, host cells and treatment regimens may account for these differences, it is apparent that the activity of DCQ in these protracted assays does not at all correlate with the indications of potency provided by the incipient assays in vitro against *T. gondii*. 

DCQ had been previously examined in efficacy studies in vivo in different mouse models for *Plasmodium* and *Cryptosporidium* infection [34,35,36] without causing any serious adverse effects. Thus, we assessed the efficacy of DCQ in vivo in a pregnant mouse model for *T. gondii* ShSp1 oocyst infection. However, DCQ treatment of *T. gondii* infected pregnant mice at 10 mg/kg/day for 5 days resulted in a detrimental outcome. While fertility was not affected by the drug—92 pups were born from eight dams in both, the DCQ-treated and the C+ group—the rate of neonatal mortality was almost twice as high in the DCQ-treated group. In addition, all pups of the DCQ group that survived until 30 days post-partum were *T. gondii* PCR-positive, while in the C+ group 20 of 55 pups were PCR-negative. This finding is surprising, as DCQ is not known to have embryotoxic or teratogenic effects [13]. However, most, if not all, published studies employing DCQ were carried out in non-pregnant mice, and the drug is specifically marketed for the treatment of gastrointestinal coccidiosis in cattle, small ruminants and poultry. Pharmacokinetic studies in milking cows that received 0.5 mg/kg DCQ had shown that the compound was well absorbed and reached maximum plasma concentrations of 2 μM [14], which was well above the IC_50_ value for *T. gondii*. DCQ is also marketed for use in pregnant ewes for prophylaxis of *Toxoplasma*-induced abortion, with a recommended dose of 2 mg/kg during the last 14 days of pregnancy without any obvious adverse effects on pregnancy outcome. It is possible that the timing of DCQ treatment could have had an impact in this study. Teratogenic effects are most problematic for the fetus during the first trimester of development [37], during which the fetal tissue undergoes rapid cell proliferation [38]. The distinct anatomical and physiological dissimilarities between ruminant and mouse placenta could also account for the observed effects in this study [39], together with other differences, such as species-dependent bioavailability, metabolic stability, systemic exposure, and other pharmacokinetic properties. Overall, in this mouse model, DCQ treatment appeared non-efficacious, caused serious adverse effects, and led to increased neonatal pup mortality.

Of the three quinoline-*O*-carbamate derivatives of DCQ investigated in this study, dose-response assessments in vitro showed that RMB060, in particular, had highly favorable properties, with a selectivity index of >140,000 when the drug was applied concomitantly to infection, and >14,000 upon addition of the compound to intracellular parasites at 3 h post-infection. The previously reported IC_50_ for RMB060 was 1.1 nM [18], which largely agrees with the results obtained in this study (0.07–0.7 nM, depending on the timing of treatment). The IC_50_ value of RMB054 was also in agreement with the previous report. However, results on RMB050 are clearly different, with an IC_50_ value that is over 20 times higher than reported earlier [18]. Currently, we have no explanation for this discrepancy.

Ultrastructural investigations on the impact of RMB060 treatment on *T. gondii* tachyzoites cultured in HFF were carried out by treating infected cultures with 500 nM of the compound, which corresponds to the concentration that caused a complete growth inhibition in dose response assays. This showed that the compound primarily induced alterations in the tachyzoite mitochondrion and especially affected the matrix and the cristae. This is similar to observations reported with other drugs that target the cytochrome *bc_1_* complex in apicomplexan parasites, such as in DCQ- and buparvaquone-treated *Besnoitia besnoiti* tachyzoites [12,40], and in endochin-like quinolone-treated *N. caninum* tachyzoites [41]. Interestingly, despite the low IC_50_ values determined in short-term growth assays (0.07–0.7 nM), exposure to the drug for longer timespans showed that *T. gondii* tachyzoites readily adapted to concentrations that were up to 4286-fold the IC_50_, resumed proliferation (albeit at a lower rate than non-treated parasites) and formed lysis plaques. This underlines the impressive adaptive potential of *T. gondii*, as seen previously with other drugs, such as dicationic pentamidine derivatives [42]. On one hand, this adaptive potential could be based on true resistance formation; for example, point mutations in the cytochrome *b* gene, as seen in buparvaquone resistant *Theileria* [43]. On the other hand, adaptation to increased drug concentrations could also be based on an altered gene expression pattern, such as increased expression of genes coding for components of the *Toxoplasma* detoxification machinery, e.g., ATP-binding cassette (ABC) transporters, which represent an important family of membrane proteins involved in drug resistance and other biological activities [44]. Alternatively, parasites develop an enhanced adaptation to oxidative stress.

Interestingly, although we found that treatment with RMB060 in vitro induced the expression of the bradyzoite marker BAG1, the TEM of parasite cultures submitted to long-term treatment with RMB060 did not reveal the presence of additional structural features of bradyzoites. Thus, no differentiation into cyst-forming bradyzoites took place. Nevertheless, the alterations in the mitochondrial matrix also appeared to be persistently present in drug-adapted parasites undergoing prolonged treatment with each of RMB060 and DCQ. We hypothesize that the dissolution of the mitochondrial matrix and the cristae could constitute an escape mechanism for the parasites to overcome the detrimental effects of the drug that acts on the cytochrome *bc_1_* complex. In the presence of intact cristae, this would induce free radical and ROS formation, resulting in a parasitostatic, but not parasiticidal, effect. Since *T. gondii* tachyzoites which exhibit such altered mitochondria are fully viable, their energy metabolism is most likely dependent on glycolysis. Moreover, the intermediate metabolites obtained from intact mitochondria must be scavenged from the host cell. Certainly, more research is needed to elucidate this mechanism that leads to this marked tolerance to increased drug concentrations. 

In terms of drug efficacy in vivo, a parasitostatic effect could be an advantage. If the parasite is kept at a subclinical level, the antigenic stimulus might still be sufficiently high to provide a sustained immune response, potentially preventing reinfection after recovering from the acute stage, as it has been postulated for other compounds, such as bumped kinase inhibitors (BKIs) [45,46]. We thus assessed the efficacy of RMB060 in the pregnant *T. gondii* ShSp1 oocyst infection model. However, although the treatment of *T. gondii*-infected pregnant mice at 10 mg/kg/day for 5 days with RMB060 showed better results than DCQ, the outcome was not as expected based on the promising in vitro efficacy. The fertility was not affected by the drug—125 pups from 10 dams were born in the RMB060 group, and 113 pups from nine dams in the C+ group. Additionally, neonatal and postnatal mortality were lower in the RMB060 treatment group—16.8% and 3.8% compared to 23% and 2.9% in the C+ group, respectively. Likewise, treatment reduced the cerebral parasite burden as assessed by the number of *T. gondii* tachyzoites per µg DNA in all groups, including non-pregnant mice, dams and pups. Nevertheless, vertical transmission could not be prevented and all pups from *T. gondii* PCR-positive dams that survived until 30 days post-partum tested PCR-positive for *T. gondii* in the brain in both the RMB060-treated and the C+ groups. Overall, in this mouse model, RMB060 showed a tendency to be active against *Toxoplasma* infection in non-pregnant mice and dams, though vertical transmission could not be prevented. Finally, it must be pointed out that definitive evaluation of the effects of RMB060 must await pharmacokinetic studies of the compound. It is structurally related to the *O*-quinoline carbonate ELQ-337 prepared from the antimalaria-active quinolone ELQ-300, wherein the derivative acts as a prodrug for the quinolone. Upon oral administration in a mouse model, the parent drug ELQ is formed without detectable persistence of the intact prodrug in murine plasma; however, greatly enhanced levels of ELQ-300 are observed through the administration of ELQ-337 [21]. Likewise, the quinoline-*O*-carbonate prodrug ELQ-334 has been prepared from the quinolone ELQ-316, which is active against the apicomplexan parasite *Neospora caninum*—here also, the application of the prodrug results in enhanced plasma levels of the parent quinolone [47].

## 4. Materials and Methods

### 4.1. Cell Culture Equipment and Media, Biochemicals, and Compounds

Unless stated otherwise, all cell culture devices were purchased from Sarstedt (Sevelen, Switzerland), media from Gibco-BRL (Zürich, Switzerland) and biochemicals from Sigma (St. Louis, MO, USA). DCQ was kindly provided by Prof. Gilles Gasser, Chimie ParisTech—PSL, University of Paris. RMB054, RMB055, and RMB060 were synthesized and purified as previously described [18]. All compounds were received as powder. For in vitro studies, stock solutions of 1 and 10 mM of the compounds in dimethyl-sulfoxide (DMSO) were prepared and stored at −20 °C.

### 4.2. Host Cells and Parasites

Human foreskin fibroblasts (HFF; PCS-201-010^TM^) were cultured as described [45]. Tachyzoites of *T. gondii* β-gal (Tg-β-gal) constitutively expressing β-galactosidase were kindly provided by Prof. David Sibley, Washington University, St. Louis, USA; the *T. gondii* Me49 (TgMe49) strain was obtained from Dr. Furio Spano, Istituto Superiore di Sanita, Rome, Italy; a low-passage number Spanish *T. gondii* Pig isolate (TgPgSp1) was obtained from SALUVET-Innova (ETCU-UCM), Madrid [48]. They were maintained as previously described [46]. *T. gondii* oocysts of the type II isolate TgShSp1 [49] were received from SALUVET, Complutense University of Madrid, Spain, and stored at 4 °C until use.

### 4.3. Cytotoxicity, Anti-T. gondii Efficacy Assessments and Long-Term Drug Treatment In Vitro

HFF cytotoxicity assessments and IC_50_ determinations using cultures of Tg-β-gal tachyzoites grown in HFF were carried out as described previously [18,50]. IC_50_ values were calculated with the logit-log-transformation of the relative growth and subsequent regression analysis was done with the corresponding software tool contained in Microsoft Excel software package (Microsoft, Seattle, WA). For the long-term treatments, HFF grown in T25 culture flasks were infected with 5 × 10^5^ TgMe49 tachyzoites, and after 4.5 h 0.5 μM DCQ, RMB054 or RMB060 were added. Infected cultures treated with the highest corresponding concentration of DMSO (0.5%) served as controls. Prolonged treatments for up to 24 days were carried out, with the addition of fresh medium containing the respective drug every 3–4 days. Cultures were inspected by light microscopy on a daily basis. On days 3, 6, 9, 13, 16, 20 and 24, the compound-containing medium was removed, and parasites were maintained without drug pressure until host cell lysis (plaque formation) was evident. 

### 4.4. Microscopy

For transmission electron microscopy (TEM), HFF grown in T25 culture flasks were infected with 10^7^ TgMe49 tachyzoites and incubated for 3 h at 37 °C and 5% CO_2_. Subsequently, cultures were exposed to continuous treatment of 6h, 24 h, 48 h, and 72 h with 0.5 μM DCQ, RMB054 or RMB060, and 1 µM RMB055. Infected cultures treated with DMSO served as controls. TEM preparations were performed as described earlier [46,50,51]. Samples were observed on a Philips CM 12 (Philips Research, Eindhoven, Holland) or a FEI Tecnai Spirit BioTwin TEM (TESCAN GmbH, Dormund, Germany), both operating at 80 kV.

For immunofluorescence microscopy, 4 × 10^4^ HFF were grown on glass coverslips for 24 h to form a sub-confluent monolayer and were exposed to 1.2 × 10^5^ TgMe49 tachyzoites during 4 h at 37 °C and 5% CO_2_. The medium was replaced and supplemented with 20 nM DCQ and RMB060, or medium containing DMSO as control. Cultures infected with TgPgSp1 (low passage number) served as positive control for BAG1 (bradyzoite antigen 1) expression. Infected cells were cultured for 16 h, 24 h, and 48 h and fixed and labelled for immunofluorescence staining as described previously [50,51]. A polyclonal rabbit anti-BAG1 antiserum was applied at 1:200 dilution, followed by a goat-anti-rabbit fluorescein-isothiocyanate conjugate (1:300), the monoclonal mouse anti-TgSAG1 antibody was diluted (1:100), followed by goat anti-mouse conjugated to tetramethylrhodamine-isothiocyanate at 1:300. Subsequently, samples were mounted in H-1200 Vectashield mounting medium (Vector Laboratories, Burlingame, CA, USA) containing 4,6-diamidino-2-phenylindole and were viewed on a Nikon E80i fluorescence microscope (Tokyo, Japan). Processing of images was performed using Openlab 5.5.2 (Improvision, PerkinElmer, Waltham, MA, USA) and the ImageJ software, version 1.53 m.

### 4.5. Ethics Statement

Animal experiments were approved by the Animal Welfare Committee of the Canton of Bern under the license BE117/20. CD1 and BALB/c mice were purchased from Charles River (Sulzberg, Germany) at 6 weeks of age, and were maintained in a common room under controlled temperature and a 14 h/10 h dark/light cycle, with food and water ad libitum. Prior to the experiment, they were housed for two weeks in the facility for adaptation. The animals were handled in strict accordance with practices to minimize suffering.

### 4.6. Pregnancy Interference Test in BALB/c Mice 

Eighteen female and 9 male BALB/c mice, all 8 weeks of age, were used. The females were oestrus-synchronized for 3 days by the Whitten effect and were mated by housing 2 females and 1 male for 72 h in one cage. Males were removed after mating and females were randomly assigned to 3 experimental groups of 6 mice each: DCQ; RMB060; placebo. On days 9–13 post-mating, DCQ and RMB060 were supplied at 10 mg/kg/day in 100 μL corn oil by oral gavage. The placebo group only received corn oil. Mice were observed and weighed on several days and on day 18, post-mating pregnant females were separated into single cages. Dams gave birth between days 20 and 22. Dams and pups were observed daily and data on fertility, litter size, neonatal and postnatal mortality was recorded for the following 2 weeks. All mice were euthanized in a chamber by isoflurane/CO_2_ inhalation.

### 4.7. DCQ Efficacy Assessment in CD1 Mice Orally Infected with TgShSp1 Oocysts

Twenty-nine female and 15 male CD1 mice, all 8 weeks of age, were used. Oestrus-synchronization and mating was carried out as described under Section 4.6. Subsequently, females were distributed randomly to 3 experimental groups: DCQ, DCQ + infection (*n* = 12); C+, corn oil + infection (*n* = 12); C-, only corn oil (*n* = 5). Infection was performed with 100 TgShSp1 oocysts at day 7 post-mating by oral gavage; the C− group received PBS. The treatment with DCQ suspended in corn oil at 10 mg/kg/day for 5 days; that only with corn oil was initiated at 2 days post-infection. At day 18 post-mating, pregnant mice were distributed into single cages and non-pregnant mice were housed together with 3–4 animals per cage. Dams gave birth between day 20–22, and clinical status, litter size, neonatal and postnatal mortality were recorded daily. Four weeks post-partum (p.p.), all mice were euthanized in a chamber by isoflurane/CO_2_ inhalation. Blood and brain samples were collected. Total IgG was measured in *T. gondii* infected mice by ELISA [49].

### 4.8. RMB060 Efficacy Assessment in CD1 Mice Orally Infected with TgShSp1 Oocysts

Eight-week-old CD1 mice, 29 females and 15 males, were used. After oestrus-synchronization and mating, females were randomly distributed to 3 experimental groups: RMB060, RMB060 + infection (*n* = 12); C+, corn oil + infection (*n* = 12); C−, only corn oil (*n* = 5). Infection was carried out at day 7 post-mating with 100 TgShSp1 oocysts suspended in 100 µL PBS, with the C−group receiving only PBS. Treatments, separation of pregnant and non-pregnant mice, and monitoring were performed as described in Section 4.7. All mice were euthanized 4 weeks p.p. by isoflurane/CO_2_. Blood and brain samples were collected. Total IgG was measured in *T. gondii* infected mice by ELISA to see whether drug treatment would have an impact on the humoral immune response [49].

### 4.9. Determination of Cerebral Parasite Load by Real-Time PCR

Quantification of the cerebral parasite load in non-pregnant mice, dams and pups was carried out with an established real-time PCR method for the detection of *T. gondii* [52]. The NucleoSpin DNA RapidLyze Kit (Macherey-Nagel, Oensingen, Switzerland) was used for DNA purification according to the standard protocol and DNA concentrations were quantified by the QuantiFluor double-stranded DNA (dsDNA) system (Promega, Madison, WI, USA). Quantitative real-time PCR was performed with the Light Cycler (Roche, Basel, Switzerland), and the parasite load was calculated by including a standard curve of DNA samples from 1000, 100, and 10 *T. gondii* tachyzoites per run [50,53].

### 4.10. Statistical Analysis

Graph creation and statistical analysis was performed in GraphPad Prism version 5.0 (GraphPad Software, La Jolla, CA, USA). Cerebral parasite burdens were compared between groups by the non-parametric Kruskal–Wallis test, followed by Mann–Whitney U test. Pup mortality over time was depicted by plotting survival events at each time point in Kaplan–Meier graphs and survival curves were compared by the Log-rank (Mantel–Cox) test.

## 5. Conclusions

In conclusion, DCQ and its derivatives are very active against *T. gondii* in vitro. However, results of the experiments in vivo were disappointing in that treatment with DCQ did not clear parasitaemia and had detrimental effects on pup survival. Although RMB060 reduced infection in adult mice, vertical transmission could not be prevented. Overall, whilst rapidly proliferating tachyzoites are readily expunged by the drugs as indicated by the short-term growth assays in vitro, long-term exposure results in growth inhibition, but no parasiticidal activity is detected, indicating that *T. gondii* is able to adapt to drug pressure. The outcome of the current work illustrates the present need for the rational development of drugs, and especially drug combinations, that can target the quiescent tachyzoites and subsequent stages.

## Figures and Tables

**Figure 1 molecules-26-06393-f001:**
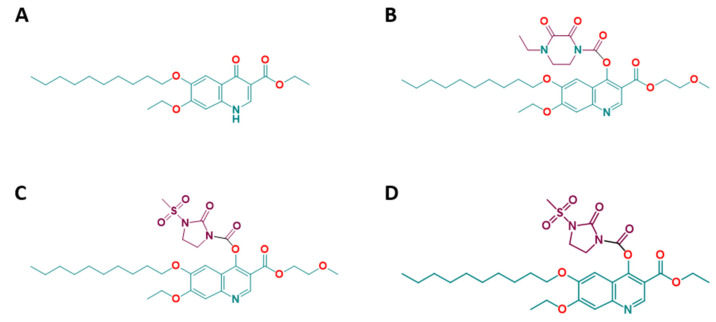
Structures of decoquinate (DCQ) and its quinoline–*O*-carbamate derivatives. (**A**) DCQ, C_24_H_35_NO_5_, MW = 417.546; (**B**) RMB054, C_32_H_45_N_3_O_9_, MW = 615.724; (**C**) RMB055, C_30_H_43_N_3_O_10_S, MW = 637.745; (**D**) RMB060, C_29_H_41_N_3_O_9_S, MW = 607.719.

**Figure 2 molecules-26-06393-f002:**
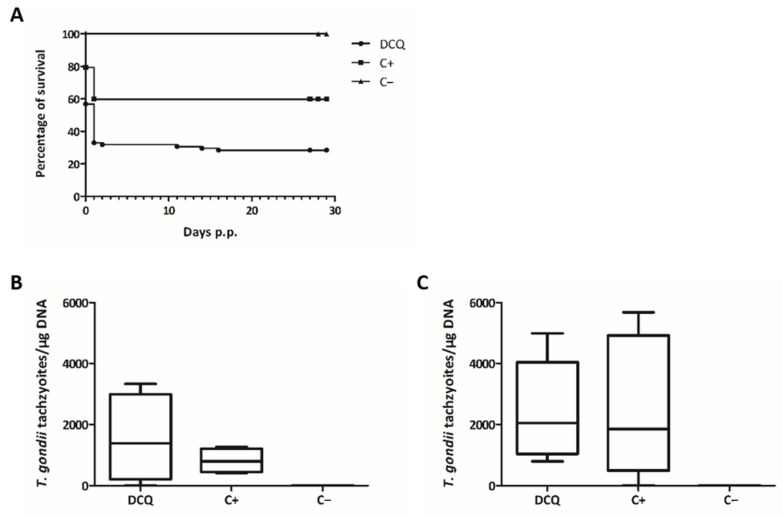
DCQ treatment in the pregnant toxoplasmosis mouse model. Survival curves of pups (**A**) and cerebral parasite burden of non-pregnant mice (**B**) and dams (**C**). Survival rates were plotted in Kaplan–Meier graphs at each time point and curves were compared by the Log-rank (Mantel–Cox) test. Differences between the three curves were highly significant (*p* < 0.0001). CD1 mice were infected with 50 TgShSp1 oocysts and treated with 10 mg/kg DCQ, while C+ was infected and treated only with corn oil. C− was not infected and treated with corn oil alone. Four weeks p.p. all mice were euthanized, brains were aseptically collected, and cerebral parasite burden was quantified by real-time PCR. The values are depicted as box plots and parasite burden between groups were compared by the Kruskal-Wallis test, followed by the Mann–Whitney U test. No statistically significant differences in the cerebral parasite burden were observed between DCQ-treated mice compared to the C+ group (neither in non-pregnant mice nor in dams).

**Figure 3 molecules-26-06393-f003:**
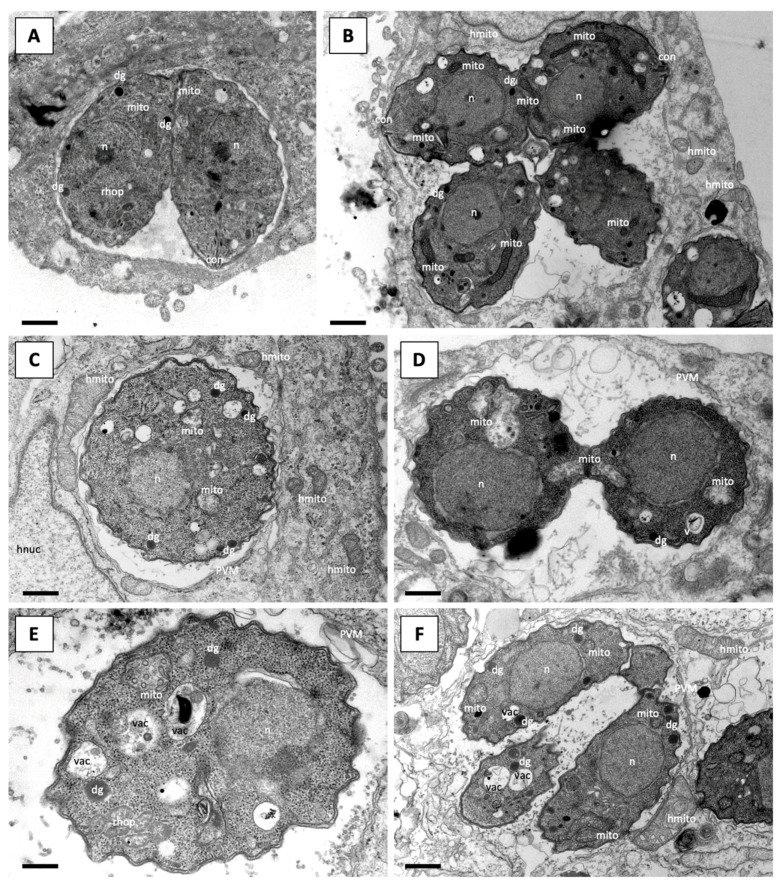
TEM of *T. gondii* Me49 tachyzoites cultured in vitro in HFF host cells. HFF monolayers were infected with *T. gondii* tachyzoites and drug treatment was initiated 3 h later. (**A**,**B**) show TgMe49 treated with DMSO as control for 6 h and 24 h, respectively. TgMe49 treated with 0.5 µM RMB060 for 6 h are shown in (**C**), 24 h in (**D**), 48 h in (**E**) and 72 h in (**F**). con = conoid; dg = dense granules; mito = mitochondrion; hmito = host cell mitochondrion; n = nucleus; hnuc = host cell nucleus; PVM = parasitophorous vacuole membrane; rhop = rhoptries; vac = cytoplasmic vacuole. Bars in (**A**) = 1 µm; (**B**) = 1.1 µm; (**C**) = 0.8 µm; (**D**) = 0.9 µm; (**E**) = 0.5 µm; (**F**) = 1.2 µm.

**Figure 4 molecules-26-06393-f004:**
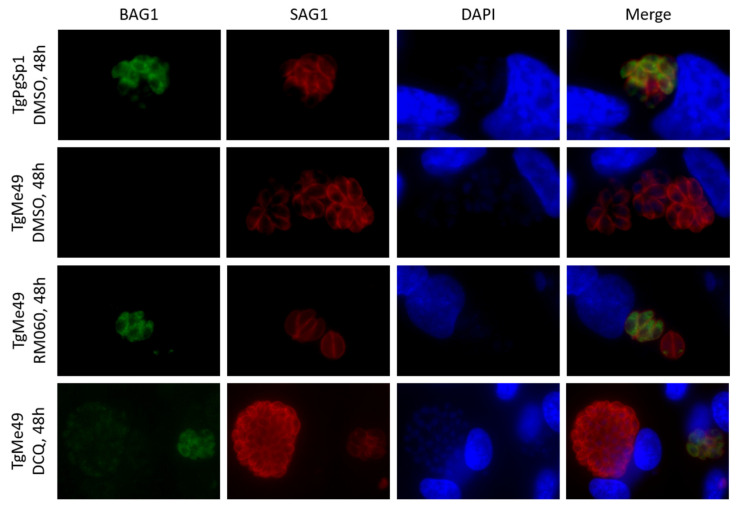
Immunofluorescence staining of HFF monolayers infected with *T. gondii* Me49 (TgMe49) tachyzoites after treatment with RMB060 and DCQ for 48 h. TgMe49 treated with the corresponding amount of DMSO were used as negative control. As positive control, TgPgSp1 were fixed and processed identically. BAG1 is labelled in green, SAG1 staining is red, and nuclei are stained with DAPI.

**Figure 5 molecules-26-06393-f005:**
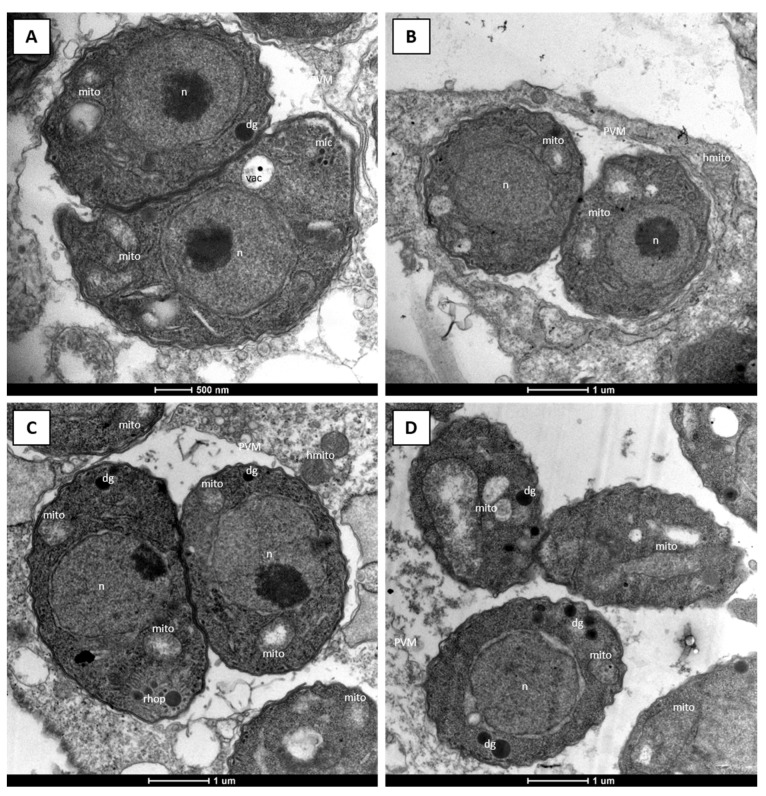
TEM of *T. gondii* Me49 tachyzoites cultured in vitro in HFF host cells. Long-term treatment with increasing concentrations of RMB060 and DCQ over 52 days, with end concentrations of 0.3 µM and 0.5 µM, respectively. (**A**,**B**) show TgMe49 treated with RMB060 and TgMe49 treated with DCQ are shown in (**C**,**D**). dg = dense granules; mic = micronemes; mito = mitochondrion; hmito = host cell mitochondrion; n = nucleus; hnuc = host cell nucleus; PVM = parasitophorous vacuole membrane; rhop = rhoptries; vac = cytoplasmic vacuole.

**Figure 6 molecules-26-06393-f006:**
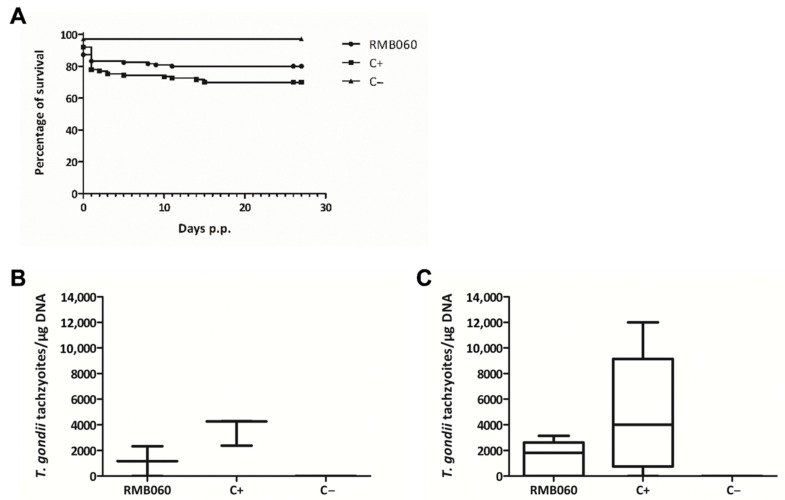
RMB060 treatment in the pregnant toxoplasmosis mouse model. Survival curves of pups (**A**) and cerebral parasite burden of non-pregnant mice (**B**) and dams (**C**). Survival rates were plotted in Kaplan–Meier graphs at each time point and curves were compared by the Log-rank (Mantel-Cox) test. No statistically significant differences were observed between the survival curves of the RMB060 group and the C+ group. CD1 mice were infected with 100 TgShSp1 oocysts and treated with 10 mg/kg RMB060, while C+ was infected and treated only with corn oil. C− was not infected and treated with corn oil alone. Four weeks p.p., all mice were euthanized, brains were aseptically collected, and cerebral parasite burden was quantified by real-time PCR. The values are depicted as box plots and parasite burden between groups were compared by the Kruskal–Wallis test, followed by the Mann-Whitney U test. No statistically significant differences in the cerebral parasite burden were observed between RMB060-treated mice compared to the C+ group (neither in non-pregnant mice nor in dams).

**Table 1 molecules-26-06393-t001:** In vitro activities against *T. gondii* and cytotoxicity against HFF.

Compounds	IC_50_ *T. gondii* (nM) [LS; LI] ^a^ Compound Added Prior to Infection	IC_50_ *T. gondii* (nM) [LS; LI] ^a^ Compound Added 3 h Post-infection	IC_50_ HFF (µM)	Selectivity Index
DCQ	1.1 [3; 0.4]	4.7 [10.1; 2.1]	>5 ^b^	>4545
RMB054	11.5 [16.6; 8]	11.4 [15.5; 8.5]	>5 ^b^	>435
RMB055	199.6 [234.6; 169.8]	226.7 [240.9; 213.4]	>10 ^b^	>50
RMB060	0.07 [0.14; 0.03]	0.7 [1.22; 0.4]	>10 ^b^	>142,857

^a^ Values at 95% confidence interval (CI); LS (limit superior) and LI (limit inferior) are the upper and lower limits of the CI, respectively. ^b^ Values cannot be calculated as there is no decline in viability over the concentration range.

**Table 2 molecules-26-06393-t002:** Litter size, parasite burden, neonatal and postnatal mortality rates of *T. gondii* infected mice treated with DCQ.

Treatment	Challenge	Seropositive for *T. gondii*	*T. gondii* Positive Non-Pregnant Mice	*T. gondii* Positive Dams	No. of Pups/No. of Dams	Neonatal Mortality	Postnatal Mortality	*T. gondii* Positive Pups
10 mg/kg DCQ	TgShSp1 oocysts	11/12	3/4	8/8	92/8	64/92	3/28	25/25
Corn oil	TgShSp1 oocysts	9/12	4/4	7/8	92/8	37/92	0/55	35/55
Corn oil	PBS	0/5	0/2	0/3	41/3	0/41	0/41	0/41

**Table 3 molecules-26-06393-t003:** Potential pregnancy interference of DCQ and RMB060 in embryo development.

Treatment	No. of Pregnant Mice/No. of Mice	No. of Non-Pregnant Mice/No. of Mice	Litter Size	Neonatal Mortality	Postnatal Mortality
10 mg/kg DCQ	2/6	4/6	7	0/7	0/7
10 mg/kg RMB060	4/6	2/6	27	3/27	0/24
Corn oil	4/6	2/6	19	1/19	0/18

**Table 4 molecules-26-06393-t004:** Litter size, parasite burden, neonatal and postnatal mortality rates of *T. gondii* infected and RMB060 treated mice.

Treatment	Challenge	Seropositive for *T. gondii*	*T. gondii* Positive Non-Pregnant Mice	*T. gondii* Positive Dams	No. of Pups/No. of Dams	Neonatal Mortality	Postnatal Mortality	*T. gondii* Positive Pups
10 mg/kg RMB060	TgShSp1 oocysts	8/12	1/2	7/10	125/10	21/125	4/104	62/100
Corn oil	TgShSp1 oocysts	10/11	3/3	8/9	113/9	26/113	8/87	70/79
Corn oil	PBS	0/5	0/3	0/2	35/2	1/35	0/34	0/34

## Data Availability

Data are made available as Appendix A (see above).

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
