# Peer review of "Assessment of the Activity of Decoquinate and Its Quinoline-O-Carbamate Derivatives against Toxoplasma gondii In Vitro and in Pregnant Mice Infected with T. gondii Oocysts"

_molecules, 2021, doi:10.3390/molecules26216393_

Round 1

Reviewer 1 Report

The paper aggregates data regarding the effect of Decoquinate and 3 of its derivatives RMB054, RMB055 and RMB060 on Toxoplasma gondii. The in vivo assays are innovative, focusing on the vertical transmission of the parasite and its effects on mortality. Another aspect to consider in the merits of the work is the repositioning of drugs developed for other diseases and tested for toxoplasmosis and its importance in vertical transmission (little explored). Although the results have not been very promising, they open perspectives for exploring this repositioning of drugs and draw attention to the activation of the gene for bradyzoites without actually presenting morphological characteristics of the tachy-bradi conversion.
The article brings new insights into the efficacy of RMB060 against T. gondii oocyst infection in pregnant mice in comparison with DCQ.

Author Response

Thank you for your positive assessment of our work. No specific points need to be addressed.

Reviewer 2 Report

Toxoplasmosis is a worldwide infectious disease that has no proper clinical treatment. Quinolones are a group of promising compounds since they were reported to be active against a wide array of bacteria and parasites. In this context, authors report herein the efficacies of DCQ and three particular quinoline-O-carbamate derivatives (named RMB055, RMB054 and RMB060) against T. gondii tachyzoites. In vitro studies with human foreskin fibroblasts (HFF) and with murine models were performed. The manuscript is well structured and well written and the main conclusions clearly come out from the results shown in the manuscript. Despite some questions still remaining unknown, this work represents a valuable contribution shedding some light and better addressing the information available in the literature regarding the potential use of these quinolones.

Below authors will find a list of some remarks that might contribute to improving the overall work.

Major remarks:

1) In Supplementary file 1, plot B the curve starts from about 60% with respect to the control experiment. The effect at lower concentration of RMB054 should be depicted in the graph. In addition, the fit curve resulting from the data analysis should replace the current lines connecting the data points.

2) Page 3 (line 119) Authors claim that the efficacy of DCQ changes depending on the drug administration procedure (i.e., prior or after infection). However, this is not so clear when looking at the data depicted in Supplementary file 1 and LS and LI values listed in Table 1. The difference seems to be clear only for compound RMB060. To clarify this, authors should provide the standard deviation for each IC50 value.

3) In Ref [18] the activity of the four investigated compounds against T. gondii in infected HFF cell lines were determined. Although in the discussion session this is mentioned, authors should further discuss their own results in the framework of the previous findings.

4) The cytotoxicity of the four compounds on HFF should be properly measured. To this end, a larger drug concentration range should be tested for each drug. This is a key aspect for any further conclusion since drugs were tested in murine models. Briefly, considering that the molecular weight of decoquinate is 417.54, and average weight of mouse of 25 gr, and considering 80 ml/Kg of total blood volume in mice... then, the dose used for oral administration (10 mg/Kg/day) is equal to a final concentration of ~300 uM per day. When considering the average volume used for oral administrations (20 ml/Kg maximum) the local concentration right upon the administration would be of ~1.2 mM per dose. The latter values are far beyond the tested range in Table 1 or Supplementary file 1.

5) In addition, why was oral administration chosen? Is there any evidence of no chemical alteration of the drugs when passing through the gastrointestinal route? The negative effect observed in pregnant mice due to the tested compounds or to any of the chemical products that might be formed upon oral ingestion?

6) (a) Figure 3 shows that as long as the incubation time with RMB060 increases a clear mitochondrial damage in T. gondii is observed. Regarding the host cell authors say “… while the host cell mitochondria remained structurally intact (Fig. 3C) (page 7, lines 195-196)”. However, an effect seems to be observed as long as the incubation time is increased (Figures 3C to 3F). Moreover, in control Figures 3A and 3B hmit are not properly labeled or shown. This state needs to be properly addressed. (b) Why was 500 nM chosen as the concentration to be tested? This should be further discussed.

9) In page 8 line 226 says “… RMB054 treated cultures, BAG1 expression was not clearly detectable (data not shown).” This information should be provided as supplementary material.

Minor comments:

- In order to make the reading easier and for a direct comparison, the calculated selectivity index values should be included in Table 1, in a separated column.

- The quality (and/or resolution) of the graph should be improved.

- Also, the size of the letters, numbers, symbols and labels within each plot should be increased in order to facilitate the interpretation. For example, it is hard to distinguish the symbols in Figure 2A and 6A.

Author Response

Responses to reviewer 2

Thank you for reviewing our manuscript and providing advice on improvement. Please find our point-by-point response below:

Major remarks:

  • In Supplementary file 1, plot B the curve starts from about 60% with respect to the control experiment. The effect at lower concentration of RMB054 should be depicted in the graph. In addition, the fit curve resulting from the data analysis should replace the current lines connecting the data points.

Response: We investigated a concentration range of 10-6 to 102 µM, which is a rather extensive range. Although it is true that for RMB054 the 10-6 µM value is only reaching 60% of the control. However, assessing lower concentrations will not have an impact on the overall IC50 value. We also don’t believe that a fit curve is necessary, and we propose to use the current lines that connect the data points. We have done this in most other publications.

  • Page 3 (line 119) Authors claim that the efficacy of DCQ changes depending on the drug administration procedure (i.e., prior or after infection). However, this is not so clear when looking at the data depicted in Supplementary file 1 and LS and LI values listed in Table 1. The difference seems to be clear only for compound RMB060. To clarify this, authors should provide the standard deviation for each IC50 value.

Response: We agree that this is not entirely clear for DCQ, and thus modified the statements on lines 119-120 (Results), and 319-323 (Discussion).

  • In Ref [18] the activity of the four investigated compounds against T. gondii in infected HFF cell lines were determined. Although in the discussion session this is mentioned, authors should further discuss their own results in the framework of the previous findings.

Response: Good point. We commented on this aspect in lines 362-367 in the discussion section.

  • The cytotoxicity of the four compounds on HFF should be properly measured. To this end, a larger drug concentration range should be tested for each drug. This is a key aspect for any further conclusion since drugs were tested in murine models. Briefly, considering that the molecular weight of decoquinate is 417.54, and average weight of mouse of 25 gr, and considering 80 ml/Kg of total blood volume in mice... then, the dose used for oral administration (10 mg/Kg/day) is equal to a final concentration of~300 uM per day. When considering the average volume used for oral administrations (20 ml/Kg maximum) the local concentration right upon the administration would be of ~1.2 mM per dose. The latter values are far beyond the tested range in Table 1 or Supplementary file 1.

Response: Cytotoxicity of these compounds has been assessed in previous work reported in reference 18, and we believe that these values and those we report here are sufficiently informative. In addition, the additional comment made by reviewer 2 is unclear to us, as we do not know where the “average volume used for oral administrations (20 ml/Kg maximum) comes from. As indicated in materials and methods, we have applied the drugs in a volume of 100 µl per mouse, and this is a standardized procedure that we have used for many other compounds in the past.

  • In addition, why was oral administration chosen? Is there any evidence of no chemical alteration of the drugs when passing through the gastrointestinal route? The negative effect observed in pregnant mice due to the tested compounds or to any of the chemical products that might be formed upon oral ingestion?

Response: Oral administration is the least invasive and preferred route of drug administration when working with pregnant mice. In our experience, applying compounds by the intraperitoneal route is not advisable in pregnant mice, and also repeated subcutaneous or intramuscular injections are not advisable, as this induces unnecessary stress to these animals that could impact on pregnancy outcome. However, the reviewer is of course right that the compounds are possibly modified when passing through the gastro-intestinal route. This aspect is not covered in the current manuscript.

  • (a) Figure 3 shows that as long as the incubation time with RMB060 increases a clear mitochondrial damage in T. gondii is observed. Regarding the host cell authors say “… while the host cell mitochondria remained structurally intact (Fig. 3C) (page 7, lines 195-196)”. However, an effect seems to be observed as long as the incubation time is increased (Figures 3C to 3F). Moreover, in control Figures 3A and 3B hmit are not properly labeled or shown. This state needs to be properly addressed. (b) Why was 500 nM chosen as the concentration to be tested? This should be further discussed.

Response: a) Actually, Figure 3 shows clearly that a mitochondrial damage within the parasite is induced with RMB060 during the first 6-24 hours of treatment, but after 48-72 hours the mitochondrial matrix and cristae within the parasite mitochondrion had regained partial structural definition. Whether this includes also functionality is not known. This is also described in the corresponding results section. Figure 3 also shows that host cell mitochondria (hmito) do not appear to be structurally affected. These mitochondria are not visible in every section, due to the nature of TEM, which just provides an image of a single section plane in a given image. We have now properly labelled host cell mitochondria in untreated parasites in Figure 3B. b) 500 nM was chosen because this is the concentration that inhibits the proliferation of the parasites to 100% (see suppl. File 1). This information is now given in the discussion, lines 369-371.

  • In page 8 line 226 says “… RMB054 treated cultures, BAG1 expression was not clearly detectable (data not shown).” This information should be provided as supplementary material.

Response: this is done, and the information is added as an additional supplementary figure 4, while the former supplementary figure 4 is now supplementary figure 5

Minor comments:

- In order to make the reading easier and for a direct comparison, the calculated selectivity index values should be included in Table 1, in a separated column.

Response: Table 1 was modified accordingly

- The quality (and/or resolution) of the graph should be improved.

Response: resolution was improved

- Also, the size of the letters, numbers, symbols and labels within each plot should be increased in order to facilitate the interpretation. For example, it is hard to distinguish the symbols in Figure 2A and 6A.

Response: we believe the letters are large enough, but if the editor requires these changes we’ll be happy to do so

Reviewer 3 Report

The work is interesting. The methodology is sound and sufficient to fulfill the underlying objectives of the study. The results are sufficiently discussed in light of the published literature. However, some revisions are required before acceptance:

Title is not proper and must be rewritten

What was the reason for selection of these concentrations in vitro and in vivo?

Line 492. 18 female changed by “Eighteen female…”

Line 503. 29 female changed by “Twenty-nine female…”

Line 512. 4 weeks post-partum changed by “Four weeks….”

Line 516. 8 week-old changed by “Eight week-old…”

Line 519. with 100 TgShSp1 oocysts. In which volume and solvent??

Line 522. Total IgG was measured in T. gondii infected mice by ELISA [49]……Was this to confirm the infection? If not, please explain

For in vitro assay, what about the control drug?

Why did the authors not perform parasitological tests on the animals studied?

What about the toxicity profile for tested doses?

Author Response

Thank you for your review of our manuscript. Please find our point-by-point response to the comments.

Title is not proper and must be rewritten

Response: We do not agree, as the reviewer does not indicate more specific details, we decide not to change the title at this stage

What was the reason for selection of these concentrations in vitro and in vivo?

IC50 evaluations followed a standard concentrations range as done in previous publications. 500 nM was chosen for TEM studies because this is the concentration that inhibits the proliferation of the parasites to 100% (see suppl. File 1). This information is now given in the discussion, lines 369-371. For in vivo studies, 10 mg/kg was used as this did not induce negative effects in the pregnant interference assay (see Table 3).

Line 492. 18 female changed by “Eighteen female…”

Response: done

Line 503. 29 female changed by “Twenty-nine female…”

Response: done

Line 512. 4 weeks post-partum changed by “Four weeks….”

Response: done

Line 516. 8 week-old changed by “Eight week-old…”

Response: done

Line 519. with 100 TgShSp1 oocysts. In which volume and solvent??

Response: in 100 µl PBS, now mentioned on line 532-33

Line 522. Total IgG was measured in T. gondii infected mice by ELISA [49]……Was this to confirm the infection? If not, please explain

Response: this was done to see whether the drug treatment impacts on the humoral immune response. Now mentioned on line 536-537

For in vitro assay, what about the control drug?

Response: we did not include a control drug in the in vitro assays.

Why did the authors not perform parasitological tests on the animals studied?

Response: it is not clear to us what is meant by “parasitological tests”. For assessing the impact of a compound treatment, we performed real time PCR to detect and quantify brain infection, we studied the clinical signs, including vertical transmission of T. gondii, and we also looked at the serological response. We believe this is sufficient for the making a clear statement on compound efficacy.

What about the toxicity profile for tested doses?

Response: toxicity profiles are not within the scope of this publication, and toxicity evaluation is limited to two aspects. The selective index that can be calculated in vitro (parasite IC50 versus estimated HFF IC50) indicated that in vitro the drugs were rather specific and would not harm the host cells. (see Table 1). We found, however, that DCQ did have an impact on pregnancy outcome in non-infected pregnant mice, while RMB060 did not (Table 3).